# The Antidiabetic Potential of Probiotics: A Review

**DOI:** 10.3390/nu16152494

**Published:** 2024-07-31

**Authors:** Shiming Li, Zichao Liu, Qi Zhang, Dan Su, Pengjie Wang, Yixuan Li, Wenbiao Shi, Qian Zhang

**Affiliations:** 1Department of Nutrition and Health, China Agricultural University, Beijing 100193, China; shiming.li@cau.edu.cn (S.L.); liuzichao0526@163.com (Z.L.); zhangqi@cau.edu.cn (Q.Z.); wpj1019@cau.edu.cn (P.W.); liyixuan@cau.edu.cn (Y.L.); 2College of Food Science and Nutritional Engineering, China Agricultural University, Beijing 100193, China; 3Department of Chemistry and Chemical Biology, Cornell University, Ithaca, NY 14850, USA; ds796@cornell.edu

**Keywords:** T2DM, probiotics, gut microbiota

## Abstract

Diabetes has become one of the most prevalent global epidemics, significantly impacting both the economy and the health of individuals. Diabetes is associated with numerous complications, such as obesity; hyperglycemia; hypercholesterolemia; dyslipidemia; metabolic endotoxemia; intestinal barrier damage; insulin-secretion defects; increased oxidative stress; and low-grade, systemic, and chronic inflammation. Diabetes cannot be completely cured; therefore, current research has focused on developing various methods to control diabetes. A promising strategy is the use of probiotics for diabetes intervention. Probiotics are a class of live, non-toxic microorganisms that can colonize the human intestine and help improve the balance of intestinal microbiota. In this review, we summarize the current clinical studies on using probiotics to control diabetes in humans, along with mechanistic studies conducted in animal models. The primary mechanism by which probiotics regulate diabetes is improved intestinal barrier integrity, alleviated oxidative stress, enhanced immune response, increased short-chain fatty acid production, etc. Therefore, probiotic supplementation holds great potential for the prevention and management of diabetes.

## 1. Introduction

### 1.1. Diabetes Mellitus

Diabetes mellitus is a chronic metabolic disease that is characterized by high blood sugar levels and circulation insulin, leading to significant morbidity and mortality worldwide [1]. According to the International Diabetes Federation Diabetes Atlas (9th edition), approximately 463 million adults globally suffer from diabetes. This number is projected to rise to 578 million (10.2% of the total population) by 2030 [2], making diabetes one of the most serious threats to human health in the 21st century. Diabetes mellitus is characterized by elevated blood glucose levels due to defects in insulin secretion and/or action [3]. Diabetes is classified into type 1 diabetes mellitus (T1DM), type 2 diabetes mellitus (T2DM), and gestational diabetes mellitus (GDM) based on genetics, etiology, and diagnostic criteria. The most common form of diabetes is T2DM, which accounts for 90% of global diabetes cases. Therefore, T2DM is the main target of diabetes prevention and treatment [4]. The main objective of this review is to discuss the effects of probiotics on T2DM and the underlying mechanisms.

The detrimental effects of diabetes have long been recognized. The main clinical symptoms of diabetes are increased food and water intake, increased urination, and body weight loss [4]. Diabetes can cause complications in multiple organs, such as the cardiovascular system, eyes, kidneys, and nerves, severely reducing the quality of life of patients. These complications have significant impacts on health—for example, significantly increased risk of heart disease and stroke [4,5]. Recent studies showed that the incidence and mortality of cardiovascular and cerebrovascular diseases in patients with diabetes were about 3.5-fold higher than those in non-diabetic patients. Diabetes patients are prone to hypertension, coronary heart disease, and myocardial infarction [6]. Diabetes can also cause serious damage to the glomeruli, leading to proteinuria, hypertension, a gradual decline in renal function, and potentially renal failure [7]. Moreover, diabetes can cause neuritis and peripheral neuropathy, resulting in foot ulcers and necrosis and sometimes necessitating amputation in severe cases [8]. In addition, diabetes can cause retinopathy, accompanied by the risk of blindness [9]. Recent research showed that patients with diabetes are at a higher risk of being infected with the 2019 novel coronavirus (COVID-19) [10,11]. Currently, the key strategies for managing diabetes include blood sugar monitoring, diet control, and exercise to maintain blood sugar in the appropriate range.

Currently, insulin injections, oral hypoglycemic drugs, and lifestyle management are the main treatments for diabetes [12]. However, a long-term regime of insulin injection may lead to insulin resistance, which may aggravate T2DM symptoms [13]. Moreover, T2DM is often associated with insulin resistance, making direct insulin injections less effective [14]. Hypoglycemic drugs may have side effects, including gastrointestinal discomfort and allergic reactions, as well as causing tissue damage to the liver, kidney, and nervous system [15]. Therefore, many researchers have been seeking treatments that have minimal side effects and can quickly and effectively control or even cure diabetes.

Besides innate genetic genes, certain unhealthy lifestyles are closely related to diabetes, including a lack of exercise, frequent intake of a high-sugar and high-fat diet, smoking, and alcohol abuse [16,17]. Overnutrition due to ingestion of a diet rich in sugar and fat is the most common cause of the development of diabetes. Intestinal microorganisms serve as a crucial link between diet and human health. Studies have shown that the metabolites produced by the host’s intestinal microorganisms are closely related to the risk of diabetes [18]. Therefore, probiotics can be used to regulate the structure and metabolism of intestinal flora when ingested as dietary supplements, thereby managing and inhibiting the development of diabetes.

### 1.2. Gut Microbiota

Recent epidemiological, physiological, and omics findings, along with cell-based and animal experimental results, indicate that a significant portion of the environmental impact on human health and disease risk may be mediated or modified by the microbial community [19]. These microbiotas include numerous interacting bacteria, archaea, phages, eukaryotic viruses, and fungi that coexist on the surface of the human body and in all body cavities, with most of them being mutually beneficial [19]. The number of genomes of all intestinal microbial genes in an individual is more than an order of magnitude greater than the human genome. Most of the microorganisms that inhabit the human body reside in the intestine and are influenced by lifestyle, drugs, and host genetics, especially dietary feeding during infancy. The intestinal microbiota regulates host immunity [20], digestive ability [21], intestinal endocrine function [22], neural signaling [23], drug metabolism [24], and the elimination of toxins by producing a variety of compounds that affect the host [25].

Much evidence has shown that gut microbiota influences the capacity of the distal intestine to secret hormones regulating blood glucose. Patients who have undergone complete colectomy have an increased risk of T2DM compared to those who do not have the surgery [26]. Mechanistic studies in mice have shown that hyperglycemia may increase intestinal barrier permeability by disrupting the tight junction integrity of glucose transporter 2 (GLUT2)-dependent intestinal epithelial cells, leading to mucosal leakage [27]. Therefore, there is significant interest in understanding whether abnormal gut microbiota contributes to the onset or maintenance of elevated blood glucose in T2DM and its precursor states.

### 1.3. Probiotics

Probiotics are live microorganisms that provide health benefits to the host when taken as dietary supplements. Probiotics can reduce the abundance of harmful bacteria in the intestinal flora and increase the abundance of beneficial bacteria, thereby regulating intestinal metabolism. In addition, probiotics increase the integrity of the intestinal barrier, thereby alleviating intestinal inflammation and reducing the risk of pathogen infection. When administered in adequate amounts (at least 10^6^ CFU), probiotics can improve the balance of intestinal microorganisms and engage in the metabolism of the host [28]. In recent years, probiotics have been used to beneficially modulate the abundance of intestinal microbiota. According to a new classification of the genus previously known as *Lactobacillus* proposed in 2020 [29,30], new names were used when referring to the former genus *Lactobacillus* in this review.

Commonly reported probiotics include lactic acid bacteria (e.g., lactobacilli (formerly *Lactobacillus*), *Bifidobacterium*, *Streptococcus*), non-lactic acid-producing bacteria (e.g., *Bacillus*, *Propionibacterium*), non-pathogenic yeasts (e.g., *Saccharomyces cerevisiae*), and non-spore-forming and non-flagellated cocci [31]. Among them, lactobacilli and *Bifidobacterium* have been the most extensively studied extensively. Lactobacilli includes different species, among which the probiotics are *Lab. acidophilus*, *Lbs. rhamnosus*, *Lab. delbrueckii subsp. bulgaricus*, *Lmb. reuteri*, *Lbs. casei*, *Lab. johnsonii*, and *Lpb. plantarum*. *Bifidobacterium* belongs to the phylum *Actinobacteria*, with common probiotic species including *B. animalis*, *B. bifidum*, *B. breve*, *B. infantis*, *B. lactis*, and *B. longum* [32]. The ability of different probiotics to exert their biological activity in vivo depends on their specific properties, such as tolerance to acidic pH environments, resistance to digestion by bile and pancreatic juice, and high efficiency of colonization in the intestine [33]. 

Probiotics have been reported to offer numerous beneficial biological functions in the host intestine. For instance, probiotic supplementation may strengthen the junction of intestinal epithelial cells and improve the integrity of the gastric mucosal barrier, resulting in enhanced intestinal barrier function [28,34]. Moreover, probiotics can regulate intestinal motility via mutual communication between the probiotic flora in the intestine and the enteric nervous system, thereby regulating intestinal motility [35,36]. The use of specific probiotics, such as *Lacticaseibacillus (Lbs.) rhamnosus* CNCM I-3690, *Lbs. rhamnosus* GG, and *Ultrabiotique* [*Lactobacillus (Lab.) acidophilus*, *B. lactis*, *Lactiplantibacillus (Lpb.) plantarum*, and *B. breve*], can significantly reduce inflammation, improve colitis, and promote mucosal healing [37]. Probiotics [*Lbs. rhamnosus* CNCM I-3690, *Lpb. plantarum MB452*, *Lpb. plantarum* GOS42, and *Limosilactobacillus (Lmb). fermentum GOS57*] were also shown to reduce the risk of pathogen infection by enhancing the production of intestinal mucins [38,39]. In addition, probiotics (such as *Lbs. rhamnosus* JB-1, *Lbs. rhamnosus* GG, and *Lpb. plantarum* N14) activate the intestinal immune system by stimulating innate immune receptors (e.g., Toll-like receptors (TLRs) and C-type lectin receptors (CLRs)), which promotes the production of proinflammatory cytokines and stimulates macrophages to initiate phagocytosis [40]. When probiotics interact with other microbiota in the intestine, they engage in cross-feed and other interactions, thereby affecting the metabolic capacity of the host’s intestinal microbiota. 

## 2. Negative Effects of Diabetes on the Gut Microbiome

Some bacterial genera are negatively correlated with type 2 diabetes. In the gut microbiota of T2DM patients, the abundance of *Bacteroidetes*, *Bifidobacterium*, *Faecalibacterium*, *Akkermansia*, and *Roseburia* was reduced. In contrast, the abundance of *Fusobacterium*, *Ruminococcus*, and *Blautia* in the gut microbiota of T2DM patients is higher than that in healthy controls [41]. Of note, T2DM can also lead to significant decreases in the abundance of *Bacteroides* in the intestine [42,43]. Patients with T2DM have a reduced abundance of *Enterobacteriaceae*, *Bacteroides* 20_3, and *Bacteroides vulgaris* [44,45,46]. *Roseburia*, *Faecalibacterium*, lactobacilli, *Ruminococcus*, and *Blautia* belong to the phylum *Firmicutes* and are negatively affected by diabetes [42,46,47]. Patients with T2DM generally have reduced levels of *Roseburia* as compared with their healthy cohorts [48]. *R. intestinalis* is positively associated with diabetes, while *R. inulinivorans* and *Roseburia*_272 are negatively associated with diabetes [44,46,49]. Other studies have observed a decreased abundance of *Faecalibacterium* and *F. prausnitzii* in T2DM patients. *F. prausnitzii* is a type of Gram-positive bacterium that can exert anti-inflammatory effects [47,50,51]. Lactobacilli species, such as *Lab. acidophilus* and *Ligilactobacillus (Lgb). salivarius*, are positively associated with T2DM. By contrast, *Ligilactobacillus (Lgb). gasseri*, another species within the genus lactobacilli, is inversely related to diabetes [42,44].

Patients with T2DM exhibit an imbalance in the gastrointestinal microbiota, characterized by an increase in the ratio of *Firmicutes* to *Bacteroidetes* and a decrease in lactic acid-producing species of the genera lactobacilli, *Bifidobacterium*, and *Streptococcus* [52]. These microbiotas have the potential to produce short-chain fatty acids (SCFAs) such as acetate. Acetate can be converted to butyrate through a cross-feeding mechanism [44,53,54]. SCFAs such as butyrate and propionate stimulate glucagon-like peptide-1 (GLP-1), an incretin hormone that regulates postprandial insulin secretion by increasing insulin release after glucose ingestion [55,56,57]. Moreover, SCFAs can regulate intestinal gluconeogenesis and glucose absorption into the portal vein [56,58]. In addition, T2DM may lead to an increase in pathogenic bacteria, including *Enterobacteriaceae*, in the gastrointestinal tract [52]. Dysbiosis and inflammation can weaken the intestinal barrier function, thereby increasing the risk of leaky gut syndrome [52].

In animal research, mice models are usually chosen because the intestinal structure of mice is very similar to that of humans. Researchers can induce T2DM in mice to closely observe the potential causal relationship and possible mechanisms between diabetes and intestinal microorganisms [59]. Previous studies showed that germ-free mice displayed a significant increase in adiposity and insulin resistance after receiving gut microbiota transplants from diabetic mice [60]. Consistent with human studies, diabetic rats undergoing Roux-en-Y gastric bypass (RYGB) surgery had altered gut microbiota, with an increased abundance of *Bacteroidetes*, *Proteobacteria*, *Fusobacteria*, and *Actinobacteria* and reduced levels of *Firmicutes* and *Verrucomicrobia*. RYGB surgery reduced body weight and significantly improved glucose tolerance and insulin sensitivity in diabetic rats [61]. These findings are comparable to the observations in human studies. Increased intestinal permeability in T2DM mice leads to higher levels of lipopolysaccharides (LPS) in the blood circulation, contributing to the progression of obesity and insulin resistance [62,63]. Therefore, diabetes may increase the abundance of harmful bacteria in the intestinal flora to aggravate intestinal inflammation and insulin resistance. Increasing the abundance of intestinal probiotics may alleviate diabetes.

## 3. Clinical Trial Study on the Use of Probiotics to Manage Diabetes in Humans

Some strains of lactobacilli, *Bifidobacterium*, and *Streptococcus* have been reported to control blood glucose by regulating satiety signals, maintaining gut barrier integrity, and enhancing the antioxidant activity of pancreatic cells [64,65]. The gut microbiota increases insulin sensitivity through the TGR5 pathway and reduces the expression of proinflammatory cytokines (e.g., tumor necrosis factor α (TNF-α), interleukin-6 (IL-6), and interleukin-1 (IL-1)) through nuclear factor kappa-B (NF-κB), which are associated with insulin resistance and oxidative damage to pancreatic β cells [66,67,68,69,70]. Therefore, the gut microbial dysbiosis observed in patients with T2DM may contribute to decreased insulin sensitivity, reduced insulin production, and impaired glucose tolerance.

Probiotic consumption may improve several metabolic disorders caused by T2DM, including upregulating insulin secretion pathways and reducing systemic inflammation and oxidative stress [53,71,72]. Currently, there is still uncertainty as to whether the gut microbial dysbiosis observed in patients with T2DM is a cause or consequence of glycemic dysregulation. However, in clinical studies, the administration of probiotics containing bacteria from the genera lactobacilli, *Bifidobacterium*, and *Streptococcus* can reverse the gut microbial imbalance, ultimately positively improving glucose metabolism and glycemic control [53]. This review summarizes some reports from human randomized controlled trials (RCTs) focusing on the effects of probiotic supplementation on glycemic outcomes in adults with T2DM (Table 1), specifically fasting plasma glucose (FPG), fasting plasma insulin (FPI), hemoglobin A1c (HbA1c), and homeostasis model assessment of insulin resistance (HOMA-IR). In these studies, some strains, including *Lab. acidophilus*, *Lbs. casei*, *Lbs. rhamnosus*, *Lab. delbrueckii subsp. bulgaricus*, *B. breve*, *B. longum*, and *S. thermophilus*, significantly improved blood glucose metabolism and reduced inflammatory damage in T2DM patients [73]. *Lpb. plantarum* A7 had no effect on glucose metabolism but changed fatty acid metabolism in T2DM patients [74]. *Lab. acidophilus*, *Lbs. casei*, *Lab. delbrueckii subsp. lactis*, *Bifidobacterium*, *B. longum*, and *B. infantis* increased insulin production in T2DM patients [75]. Lactobacilli, *Lactococcus*, *Bifidobacterium*, *Propionibacterium*, and *Acetobacter* reduced insulin resistance and systemic inflammation in patients with T2DM [76]. As well as some other probiotics (*Lab. acidophilus* Bb12, *Lab. acidophilus* La5, *Lab. acidophilus* NCFM, *B. bifidum*, *Limosilactobacillus* (*Lmb.*) *reuteri* ADR-1/3, and *Lmb. fermentum*), they all have a significant alleviating effect on T2DM [77,78,79,80,81,82]. These anti-diabetic probiotics have great potential to become a new clinical treatment for T2DM.

## 4. Effects of Various Probiotics on Diabetes in Rodent Studies

Several studies have demonstrated that probiotics can reduce blood sugar levels to varying degrees. Specifically, *Latilactobacillus* (*Ltb*.) *sakei* OK67 [83], *Lbs. rhamnosus* CCFM0528 [84], *Lbs. paracasei subsp. paracasei* NTU 101 [85], *Lpb. plantarum* NCU116 [86], and *Lbs. casei* CCFM0412 have been shown to effectively lower blood sugar concentrations in animal models of T2DM [87]. Different probiotics employ different mechanisms for inhibiting the development of diabetes. The treatment models, mechanisms of action, action cycles, and dosages of several probiotics are summarized in Table 2. The main mechanisms by which probiotics control diabetes through regulating intestinal microbiota lie in improving intestinal barrier integrity, reducing oxidative stress, enhancing immune response, increasing SCFA production, and providing liver protection (Figure 1).

For intestinal barrier integrity, some studies have shown that *Lpb. plantarum* IS 20506 can improve intestinal permeability and reduce LPS entering the blood by increasing the levels of occludin and ZO-1, thereby strengthening the tight junctions of the intestine [88,89]. Systemic low-grade inflammation arising from immune response in T2DM mice was also reduced. Research indicates that *Lbs. rhamnosus* CCFM0528 can significantly inhibit the levels of pro-inflammatory factors TNF-α, IL-6, IL-1β, and IL-8 in T2DM mice, while increase the production of the anti-inflammatory factor IL-10 [84]. Similarly, there is also a report showing that *Ltb. sakei* Probio-65 and *Lpb. plantarum* Probio-93 reduced the abundance of harmful bacteria and consequently reduced LPS levels in blood [90]. Regarding oxidative stress, *B. animalis* 01 and *Lbs. paracasei* NL41 can significantly increase the activities of superoxide dismutase (SOD), glutathione (GSH), and catalase (CAT) and significantly decrease malondialdehyde (MDA) in the liver of T2DM rats, thereby significantly improving antioxidant capacity (TAC) [91,92,93]. This process is regulated by Toll-like receptor 4 (TLR 4) and nuclear factor erythroid 2-related factor 2 (Nrf 2) and plays a protective role against oxidative damage by alleviating redox stress [94]. G-protein coupled receptor 43 (GPR43) can be activated by the increasing level of SCFAs (such as butyrate) produced by compound probiotics (*Lpb. plantarum*, *Lab. delbrueckii subsp. bulgaricus*, *Lbs. casei*, *Lab. acidophilus*, *B. infantis*, *B. longum*, and *B. breve*). GPR43 modulates intestinal signals of glucagon-like peptide-1 (GLP-1) and peptide YY (PYY) in intestinal L cells, promotes β-cell proliferation, reduces appetite, and thus alleviates glucose tolerance, and enhances energy utilization [91,95,96,97]. Therefore, probiotics alleviate T2DM by regulating the intestinal barrier integrity, oxidative stress, immune response, and SCFA production.

### 4.1. Lacticaseibacillus rhamnosus

*Lbs. rhamnosus* GG, first isolated in 1983, is a probiotic known for its potent gastric acid resistance and affinity for intestinal cells. It is now widely used to help control blood sugar in diabetic patients [98]. In rodent studies, the daily administration of *Lbs. rhamnosus* GG (1 × 10^8^ CFU/mL) to mice for four weeks increased glucose tolerance by reducing endoplasmic reticulum stress [99]. The daily oral administration of 10^9^ CFU/mL of *Lbs. rhamnosus* HAO 9 to diabetic mice induced by a high-fat diet significantly lowered insulin levels, fasting blood sugar, and proinflammatory cytokines IL-6 and TNF-a [100]. In addition, the administration of *Lbs. rhamnosus* GG to diabetic mice reduced insulin, glycated hemoglobin, and fasting blood sugar levels and increased serum GLP-1 levels [101]. In diabetic rats, the oral administration of *Lbs. rhamnosus* BSL and *Lbs. rhamnosus* R23 reduced insulin resistance by downregulating the expression of glucose-6-phosphatase [102]. Similar results were observed using 3-month-old male zebrafish [103]. In T2DM mice, a blend of three types of probiotics containing *Lbs. rhamnosus*, *Lab. acidophilus*, and *B. bifidum* species (1.8 × 10^9^ CFU) significantly reduced hypothalamic TLR4, IL-6, and NPY and reduced the serine kinases JNK and IKK [104]. These findings highlight the potential value of these probiotics of *Lbs. rhamnosus* species for diabetes management.

### 4.2. Lacticaseibacillus paracasei

Endotoxemia, characterized by elevated levels of circulating bacterial lipopolysaccharide, has been identified as a trigger for insulin resistance in mice. Suppressing endotoxemia by probiotic supplementation is considered an effective approach [105]. Treatment with *Lbs. paracasei* subsp. *paracasei* NTU101 has been reported to reduce the risk of T2DM by increasing levels of *Bifidobacterium animalis* subsp. *lactis* 420 and improving the intestinal environment, which helps maintain intestinal barrier integrity and prevent the transfer of bacterial lipopolysaccharide into the systemic circulation [106]. Similarly, the presence of *Lbs. paracasei* subsp. *paracasei* G15 and *Lbs. casei* Q14 in the intestine has been significantly associated with reduced intestinal mucosal permeability and improved epithelial barrier function. Additionally, *Lbs. paracasei* subsp. *paracasei* BCRC12188 has been shown to reduce circulating levels of LPS and inflammatory cytokines, including IL-1β and IL-8, and may alleviate inflammatory states and pancreatic β-cell dysfunction [107]. In SD rats with T2DM, the daily oral administration of 10^10^ CFU *Lbs. paracasei* NL41 for 12 weeks reduced insulin resistance, HbA1c, glucagon, leptin, and oxidative stress [93]. Therefore, these strains of *Lbs. paracasei* genus can alleviate T2DM by regulating intestinal barrier integrity to reduce LPS and inflammatory damage.

### 4.3. Lactiplantibacillus plantarum

Among lactic acid bacteria, *Lpb. plantarum* is a facultative heterofermentative member that has been shown to have immunomodulatory and anti-inflammatory effects, as well as promote mucosal barrier integrity [108]. In a high-fat and streptozotocin-induced T2DM rat model, the oral administration of *Lpb. plantarum* SS18-5 can control body weight, reduce fasting blood glucose and insulin levels, and increase liver glycogen levels [109]. The oral administration of *Lpb. plantarum* CCFM0236 to diabetic mice not only reduced food intake, blood glucose, glycated hemoglobin, and leptin levels but also regulated serum insulin content and HOMA-IR index [110]. In diabetic mice induced by a high-fat diet, treatment with *Lpb. plantarum* Probio-093 significantly reduced body weight and improved blood glucose levels [90]. In addition, the oral administration of 4 × 10^9^ CFU of *Lpb. plantarum* HAC01 daily for 8 weeks reduced FBG, HbA1c, HOMA-IR, and OGTT-AUC in T2DM mice through AMPK and AKT pathways and increased the area of insulin-positive β cells in pancreatic islet tissue [111]. Therefore, these strains of the *Lpb. plantarum* genus have the potential to alleviate T2DM.

### 4.4. Bifidobacterium

*Bifidobacterium* are considered to be the main inhabitants of the intestinal microbiota [112]. *Bifidobacterium* can metabolize host-derived glycans such as human milk oligosaccharides and mucins [113]. Diabetic patients have lower numbers of *Bifidobacterium* and *Faecalibacterium prausnitzii* in their intestines, both of which are Gram-positive bacteria with anti-inflammatory activity [114]. *Bifidobacterium* has been reported to control the development of diabetes. In Wistar rats with high-fat diet-induced diabetes, the oral administration of *B. longum* Bb46 (1 × 10^7^ CFU/mL) for 28 days reduced fasting blood glucose, glycated hemoglobin, triglycerides, and total cholesterol [115]. The combination therapy of different *Bifidobacterium* species (including *B. longum*, *B. bifidum*, *B. infantis*, and *B. animalis*) improved insulin resistance and reduced blood glucose levels in mice [116]. A probiotic mixture (containing 3 × 10^11^ CFU/g of *B. longum*, *B. infantis*, and *B. breve*) improved insulin signaling and reduced inflammation in adipose tissue of ApoE-/- rats [117]. The administration of *B. animalis subsp. lactis* 420 (1 × 10^9^ CFU/mL) to high-fat-diet-induced diabetic rats reduced inflammatory cytokines TNF-α and IL-1β, plasminogen activator inhibitor-1 (PAI-1), and IL-6 in mesenteric adipose tissue while increasing insulin sensitivity [118]. In HFD-fed and STZ-injected T2DM mice, the oral administration of 10^9^ CFU of *B. longum* DD98 daily for 3 weeks increased butyrate levels in the intestine and decreased pro-inflammatory cytokine levels in the pancreas, thereby improving insulin resistance [119]. Inactivated *B. longum* BR-108 (3.4 × 10^12^ cells/g) increased body weight and glucose tolerance while decreasing fat tissue weight, FBG, TC, TG, and nonestesterified fatty acid in Tsumura Suzuki obese diabetes (TSOD) mice [120]. In another study on T2DM rats, the daily oral administration of *B. animalis* 01 (10^9^ CFU) for 15 weeks reduced body weight, food and water intake, FBG, OGTT-AUC, HbA1c, HOMA-IR, TC, LDL-C, LPS, TNF-α, ALT, AST, and MDA while increasing IL-10, CAT, GSH, GSH-Px, and SOD. This probiotic inhibited the development of T2DM through IRS/PI3K/AKT and Keap1/Nrf2 signaling pathway [92]. In summary, these strains of the *Bifidobacterium* genus significantly improved the development of T2DM.

**Table 2 nutrients-16-02494-t002:** T2DM management with probiotics: in vivo studies in rodents (FPG: fasting plasma glucose; FPI: fasting plasma insulin; HbA1c: hemoglobin A1c; HOMA-IR: homeostasis model assessment of insulin resistance); TC: total cholesterol; TG: total triglycerides; OGTT-AUC: oral glucose tolerance test-area under the curve; LDL-C: low-density lipoprotein-cholesterol; ALT: alanine transaminase; AST: aspartate transaminase; GSH-Px: glutathione peroxidase; SOD: superoxide dismutase; GSH: glutathione; CAT: catalase; MDA: malondialdehyde; IL: interleukin; TLR4: Toll-like receptor 4; NPY: neuropeptide Y; SCFAs: short-chain fatty acids.

Probiotics	Subjects	Model Type	Effect	Mechanism	Treatment	Ref.
*Lpb. plantarum*, *Lab. delbrueckii subsp. bulgaricus bulgaricus*, *Lbs. casei*, *Lab. acidophilus*, *B. infantis*, *B. longum*, *B. breve*	40 Wistar rats	HFD + STZ	In probiotic group, FPG and insulin resistance decreased, and total antioxidant capacity increased.	Control of T2DM by increasing GLP-1 levels and reducing oxidative stress.	5 × 10^10^ CFU/mL in water, 4 weeks	[91]
*Lbs*. *paracasei subsp. paracasei* NL41	18 Sprague Dawley (SD) rats	HFD + STZ	In probiotic group, insulin resistance, HbA1c, glucagon, leptin, and oxidative stress decreased.	N/A.	10^10^ CFU, oral administration, once per day for 12 weeks	[93]
*B. longum* DD98 and selenium-enriched *B. longum* DD98	48 C57BL/6J mice	HFD + STZ	In probiotic group, FBG, HbA1c, and insulin resistance decreased.	Probiotics increase butyrate levels in the intestine and decrease pro-inflammatory cytokine levels in the pancreas, thereby improving insulin resistance.	1 × 10^9^ CFU, oral administration, once per day for 3 weeks	[119]
Inactivated *B. longum* BR-108 (IBL)	25 Tsumura Suzuki obese diabetes (TSOD) mice	Spontaneous obesity	In probiotic group, body weight and glucose tolerance increased, fat tissue weight, FBG, TC, TG, and nonestesterified fatty acid decreased.	Probiotics absorb cholesterol and produce short-chain fatty acids inhibiting cholesterol synthesis in the liver.	IBL (3.4 × 10^12^ cells/g), 50, 100, and 150 mg/kg BW, for 30 days	[120]
*B. animalis* 01	24 Sprague Dawley (SD) rats	HFD + STZ	In probiotic group, body weight, food and water intake, FBG, OGTT-AUC, HbA1c, HOMA-IR, TC, LDL-C, LPS, TNF-α, ALT, AST, and MDA decreased, and IL-10, CAT, GSH, GSH-Px, and SOD increased.	Activation of IRS/PI3K/AKT and Keap1/Nrf2 signaling.	10^9^ CFU, oral administration, once per day for 15 weeks	[92]
*Lbs. rhamnosus*, *Lab. acidophilus* and *B. bifidum*	24 *Swiss* mice	DIO (diet-induced obesity)	In probiotic group, FBG, food intake, intestinal permeability, LPS translocation, and systemic low-grade inflammation decreased.	Probiotics significantly reduce hypothalamic TLR4, IL-6, NPY, and reduce the serine kinases JNK and IKK.	1.8 × 10^9^ CFU, once per day for 5 weeks	[104]
*Lpb. plantarum* HAC01	50 C57BL/6J mice	HFD + STZ	In probiotic group, FBG, HbA1c, HOMA-IR, and OGTT-AUC decreased, and islet insulin-positive β cell area, and butyric acid increased.	Activating AMPK and Akt pathways in the liver.	4 × 10^9^ CFU, once per day for 10 weeks	[111]
*Lpb. plantarum* Probio-093	40 C57BL/6J mice	HFD	In probiotic group, α-glucosidase, α-amylase activity, body weight, FPG, and intestinal inflammation decreased, and SCFAs increased.	Probiotics reduce the abundance of *Deferribacteria* and *Proteobacteria*, increases the abundance of *Actinobacteria* and *Bacteroidetes*, regulates the intestinal barrier, and enhances immune response.	10^8^ CFU, once per day for 8 weeks	[90]

## 5. Conclusions

This review discussed the correlation between probiotics, intestinal flora, and diabetes and the potential ability of probiotics to alleviate diabetes. Firstly, probiotics reduce the abundance of Gram-negative bacteria by regulating intestinal flora, thereby reducing LPS levels and leading to reduced immune stress; secondly, probiotics strengthen the tight junctions of the intestinal epithelial barrier, which also leads to reduced LPS levels; third, probiotics produce more beneficial short-chain fatty acids (such as butyrate, acetate, and propionate), thereby enhancing intestinal metabolism; finally, probiotics protect the liver by reducing oxidative stress. These are the main mechanisms by which probiotics alleviate diabetes. Some probiotics from *Lbs. rhamnosus*, *Lbs. paracasei*, *Lpb. plantarum*, and *Bifidobacterium* species have many experimental reports proving their anti-diabetic effects. Currently, anti-diabetic probiotic supplements are still in the clinical trial stage, and there are no mature commercial products as a treatment method. The present study may contribute to the development of probiotic supplements with anti-diabetic effects. However, the underlying mechanism of how the imbalance of intestinal microbes affects diabetes or vice versa awaits further investigations. In addition, despite the beneficial effects of probiotics on metabolic diseases, including diabetes, the side effects and health risks due to long-term intake of probiotics have not yet been fully validated.

## Figures and Tables

**Figure 1 nutrients-16-02494-f001:**
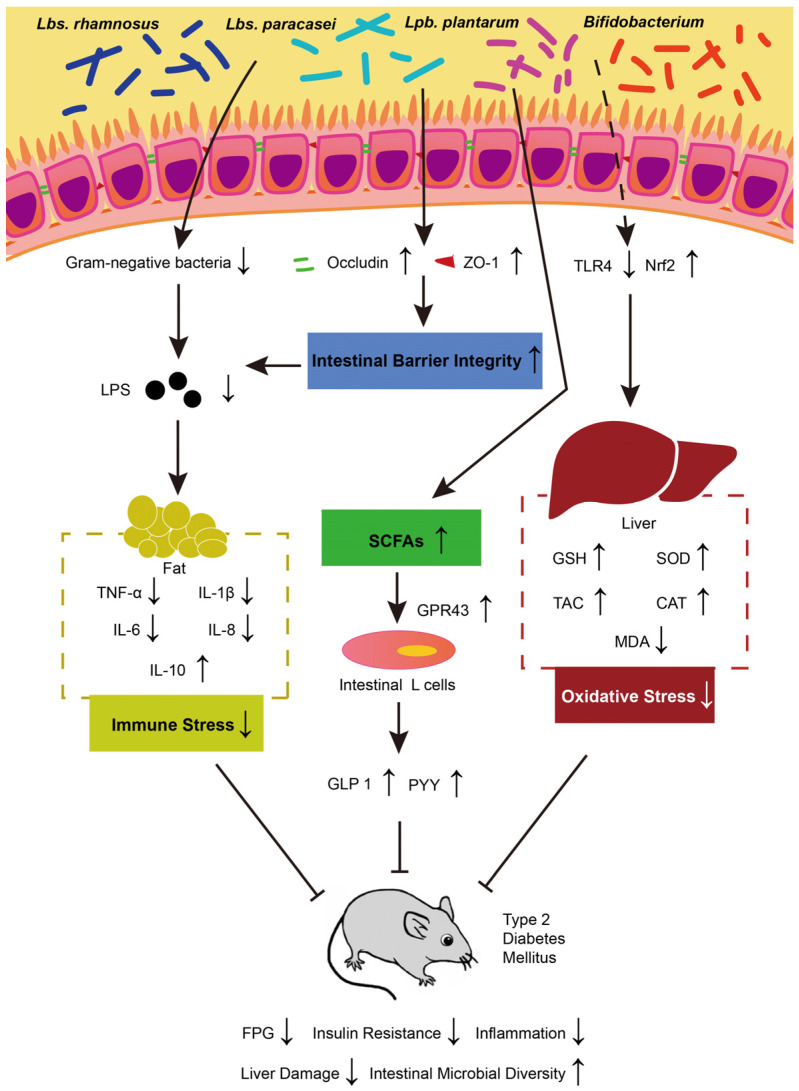
The possible mechanisms underlying probiotics’ impact on T2DM. TNF-α: tumor necrosis factor α; IL: interleukin; SCFAs: short-chain fatty acids; GLP-1: glucagon-like peptide-1; PYY: peptide YY; GPR43: G-protein coupled receptor 43; TLR 4: Toll-like receptor 4; Nrf 2: nuclear factor erythroid 2-related factor 2; SOD: superoxide dismutase; GSH: glutathione; CAT: catalase; MDA: malondialdehyde; TAC: total antioxidant capacity.

**Table 1 nutrients-16-02494-t001:** T2DM management with probiotics: studies of randomized, double-blind, controlled clinical trials using human subjects. (FPG: fasting plasma glucose; FPI: fasting plasma insulin; HbA1c: hemoglobin A1c; HOMA-IR: homeostasis model assessment of insulin resistance; hs-CRP: high-sensitivity C-reactive protein).

Probiotics	Sample	Demographics	Key Observations	Health Claim	Treatment	Ref.
*Lab. acidophilus* (2 × 10^9^ CFU), *Lbs. casei* (7 × 10^9^ CFU), *Lbs. rhamnosus* (1.5 × 10^9^ CFU), *Lab. delbrueckii subsp. bulgaricus* (2 × 10^8^ CFU), *B. breve* (2 × 10^10^ CFU), *B. longum* (7 × 10^9^ CFU), and *S. thermophilus* (1.5 × 10^9^ CFU) from the multispecies probiotic supplement (ZistTakhmir Co., Tehran, Iran) consisted of 7 viable strains.	54 Iranian adults. (n = 27)	Matched age (50.51 ± 9.82), sex, BMI (31.61 ± 6.36), and medication use.	In probiotic group, FPG and hs-CRP decreasd, and HOMA-IR and glutathione increased.	Supplementation with a multi-probiotic blend improves glucose metabolism and reduces inflammation in adults with T2DM.	Once a day for 8 weeks.	[73]
Soy milk enriched with *Lpb. plantarum* A7 (source unknown, 2 × 10^7^ CFU).	40 Iranian adults. (n = 20)	Matched age (probiotic group: 56.90 ± 1.81; control group: 53.6 ± 1.6), sex (21 male/19 female), BMI (26.68 ± 0.71), and medication use.	In probiotic group, low-density cholesterol and high-density cholesterol decreased, but fasting blood glucose did not show any significant changes.	Soy milk containing *Lpb. plantarum* A7 changes the lipid profile.	200 mL/day × 8 weeks	[74]
*Lab. acidophilus*, *Lbs. casei*, *Lab. delbrueckii subsp. lactis*, *Bifidobacterium*, *B. longum*, and *B. infantis* strains from commercial probiotics, Hexbio^®^ B-Crobes Laboratory Sdn. Bhd. (Ipoh, Malaysia), were mixed at 3 × 10^10^ CFU in water.	136 Malaysian adults (n = 68)	Matched age (52.9 ± 9.2), sex, BMI (29.2 ± 5.6), and medication use.	In probiotic group, FPI and HbA1c decreased.	Probiotics supplementation is associated with improvements in HbA1c and fasting insulin.	250 mL twice daily for 12 weeks.	[75]
Lactobacilli and *Lactococcus* (6 × 10^10^ CFU), *Bifidobacterium* (1 × 10^10^ CFU), *Propionibacterium* (3 × 10^10^ CFU), and *Acetobacter* (1 × 10^6^ CFU) from the multiprobiotic “Symbiter” (Scientific and Production Company O.D. Prolisok, Clearwater, FL, USA).	53 Ukrainian adults. (probiotic group, n = 31; control group, n = 22)	Age (probiotic group: 52.23 ± 1.74; control group: 57.18 ± 2.06). Matched BMI (34.70 ± 1.29), sex, and medication use.	In probiotic group, HOMA-IR and HbA1c decreased; chronic systemic inflammatory markers (TNF-α, IL-1β, and IL-6) decreased.	A blend of 14 probiotics reduces insulin resistance in patients with T2DM.	Once a day for 8 weeks	[76]
Probiotic yogurt enriched with *Lab. acidophilus* Bb12 (DSM 10140, 3.7 × 10^6^ CFU) and *Lab. acidophilus* La5 (Chr. Hansen, Hoersholm, Denmark, 3.7 × 10^6^ CFU).	44 Iranian adults. (n = 22)	Sex ratio: 10 male/32 female, matched age (53.00 ± 5.9), BMI (28.36 ± 4.14), and medication use.	In probiotic group, FPG, HbA1c, and TNF-α decreased.	Probiotic yogurt may be used as an alternative prevention approach and treatment method to control diabetic complications.	300 g/day × 8 weeks.	[77]
Probiotic yogurt enriched with *Lab. acidophilus* La5 (Chr. Hansen, Hoersholm, Denmark, 2.1 × 10^9^ CFU) and *B. lactis* Bb12 (DSM 10140, 1.8 × 10^9^ CFU).	60 Iranian adults. (n = 30)	Sex ratio: 23 male/41 female, matched age (51.00 ± 7.3), BMI (28.95 ± 3.65), and medication use.	In probiotic group, HbA1c and FPG decreased, and erythrocyte superoxide dismutase and glutathione peroxidase activities increased.	Probiotic yogurt improved fasting blood glucose and antioxidant status in T2DM patients.	300 g/day × 6 weeks.	[78]
*Lab. acidophilus* NCFM (ATCC 700396, Danisco Inc. (Palo Alto, CA, USA), 1 × 10^10^ CFU).	48 Danish adults. (n = 24)	Matched age (59.00 ± 6), BMI (28.1 ± 3.0), sex, and medication use.	In probiotic group, FPI decreased.	Intake of *Lab. acidophilus* NCFM preserved insulin sensitivity.	Once a day for 4 weeks.	[79]
*Lab. Acidophilic* (2 × 10^10^ CFU) and *B. bifidum* (2 × 10^10^ CFU) from Commercial probiotics, Fortitech (New York, NY, USA), and fructooligosaccharides (2 g).	20 Brazilian adults (n = 10)	All female, matched age (57.50 ± 7.5), BMI (28.2 ± 0.85), and medication use.	In probiotic group, FPG decreased, and high-density lipoprotein cholesterol increased.	This probiotic product can be used to help elderly people with T2DM maintain normal blood lipid and blood sugar levels.	Once a day for 30 days.	[80]
*Lmb. reuteri* ADR-1 (CCTCC-M207154, 4 × 10^9^ CFU) and ADR-3 (CCTCC-M209263, 2 × 10^10^ CFU).	74 Chinese adults. (n = 24–25)	Sex ratio: 38 male/36 female. Matched age (47.50 ± 32.5), BMI (28.04 ± 4.29), and medication use.	In probiotic group, HbA1c, IL-1β, and serum cholesterol decreased.	The *Lmb. reuteri* strains ADR-1 and ADR-3 have beneficial effects on T2DM patients.	Once a day for 6 months.	[81]
*Lab. acidophilus*, *B. bifidum*, *Lmb. reuteri*, and *Lmb. fermentum* (each 2 × 10^9^ CFU) from commercial probiotics, Lactocare Zisttakhmir Company (Tehran, Iran), and 50,000 IU of vitamin D3.	60 Iranian adults. (n = 30)	Matched age (71.50 ± 10.9 years), sex, BMI (29.0 ± 6.2), and medication use.	In probiotic group, FPI, HOMA-IR, and hs-CRP decreased, while serum HDL-cholesterol level, NO, TAC, and QUICKI (quantitative insulin sensitivity check index) increased.	Supplementation with probiotics and vitamin D for 12 weeks has beneficial effects on T2DM patients with coronary heart disease.	Once every 2 weeks for 12 weeks.	[82]

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
