# Peer review of "The Antidiabetic Potential of Probiotics: A Review"

_nutrients, 2024, doi:10.3390/nu16152494_

Round 1

Reviewer 1 Report

Comments and Suggestions for Authors

Thank you for giving me an opportunity to review this article. It is very interesting topic as a traditional public health issue of our time. However, there are a few things that need to be revised.

Introduction

1.     The 1st and 2nd paragraph need to be merged.

2.     From line 38 to 53, you reported that the diabetes complication in the multi-organ. However, are there other clinical problems?

3.     How about changing the order of contents; probiotics and gut microbiota? In the comprehensive introduction, you describe gut microbiota first.

4.     What benefits of probiotics are reported in this section? I looked at three advantages (those are ‘modulate the abundance of intestinal microbiota’, ‘biological functions in the host intestine’, and ‘reduce the risk of pathogen infection’). Is it right? What is the difference between modulate the abundance of intestinal microbiota and biological functions in the host intestine? Please introduce the benefits in the first sentence.

5.     Please use similar review methods; ‘3. clinical trial study ~’ and ‘4. Effects of various probiotics~’ In point ‘3’, you gave a comprehensive explanation focusing on effectiveness of probiotics. Are there all three probiotics: Lactobacillus, Bifidobacterium, and Streptococcus? However, in point ‘4’, you explained with a focus on each probiotic. Is that all? What were the criteria for selecting probiotics for this study? Please let us know.

6.     Please give us more detailed explanation or summary for table 1.

7.     In the Figure 1, some error has been shown. Immune response is reduced? Is this accurate?

8.     Please emphasize your key points at the end of each section.

Reviewer 2 Report

Comments and Suggestions for Authors

nutrients-3104834-peer-review-v1

The paper is interesting, however, in several aspects authors will need to provide additional information and to elaborate on the mechanisms that have suggested for the antidiabetic properties of the mentioned probiotics. Please, make a difference between species and strain. Probiotics properties are strain specific and not species characteristic. Maybe some additional examples and deeper description of the suggested mechanisms need to be provided and discussed. Maybe help from more experienced colleagues can be good idea in improving quality of the current manuscript.

Ln77: With the suggestions from authors of proposal for changes in taxonomy of lactobacilli it is recommended that when referring to former genus Lactobacillus from before changes in taxonomy, to use English and not Latin word - "lactobacilli"

Please, for former Lactobacillus, use new name suggested in 2020 and abbreviation according suggestion form 2023 (https://doi.org/10.1099/ijsem.0.004107.32293557 and https://doi.org/10.1163/18762891-20230114).

Ln86: Not all probiotic need to have good adherence properties. Please, discuss this section better.

Ln93-94: In this context, is better to say "strains belong to the species"

Ln97-101: Please, provide a bit more details, some examples.

Ln127-129: Please, explain better

In Table 1 will be good to provide strain numbers for all mentioned cultures or if is available, commercial name of applied probiotics.

Ln203: in this and similar occasion will be appropriate to say: Probiotic strain belongs to the species Lactiplantibacillus plantarum. And if available, provide strain identification for mentioned cultures. At the end, not all strains belong to the specific species are effective probiotics. Probiotic properties are starin and not species characteristic.

Ln 228 (and similar occasions), in the titles, please, provide full name of the species.

Ln229: Aagin, please, make a difference between species and strain. In this sentence you give impression that all strains belong to the species. L. rhamnosus are probiotics. Please, correct this. Maybe you can ask for help from more experienced colleagues to help you in the correction of the text and avoid this kind of confession.

Ln231: 8 need to be in an exponential position.

Ln232: 9 needs to be in an exponential position. Please, provide what strain of L. rhamnosus you are referring to in this section.

Ln248-249. What subsp. of L. paracasei and L. casei?

Ln228-280: maybe you can elaborate with some additional examples and more data regarding applications of strains belonging to the mentioned species as antidiabetics. Are any specific probiotics on the market with such properties? Please, elaborate in this direction into the text of the manuscript.

Concussion can be more objective. In current way is very general and not really concedes previously mentioned information’s.

Reviewer 3 Report

Comments and Suggestions for Authors

Thanks for submitting your manuscript. The topic of the manuscript is interesting, however, the novelty aspect of the manuscript is limited. There are recent literature reviews that have covered this topic. For example:

1) Probiotics and Their Role in the Management of Type 2 Diabetes Mellitus (Short-Term Versus Long-Term Effect): A Systematic Review and Meta-Analysis. https://doi.org/10.7759%2Fcureus.46741

This article is a systematic review and includes a meta-analysis based on clinical trials - three factors that make it stand out and provide strong contribution to the literature. 

2) Probiotics-based interventions for diabetes mellitus: A review. https://doi.org/10.1016/j.fbio.2021.101172

This article is similar to your manuscript in terms of theme and concept; and includes both animal and human references.

Thus, I would suggest to reject this manuscript due to lack of novelty. 

Comments on the Quality of English Language

Good - overall. 

Round 2

Reviewer 2 Report

Comments and Suggestions for Authors

nutrients-3104834-peer-review-v2

Authors have improved the text, and, in my opinion, paper can be suggested for publication. However, some adjustment of the style, and maybe proof by native English professional will be a good option for the revision of the text.

It is correct to say "lactobacilli" - without capital L and not in italics, since in this way the reference is to the English word. Please correct the entire text. Please, abbreviations for the former Lactobacillus, needs to be according to the recommendations: https://doi.org/10.1163/18762891-20230114. Please, check the entire manuscript. Example, Ln103, 275.

Reviewer 3 Report

Comments and Suggestions for Authors

No comments to authors, only to editors.

Author Response

Comments 1: No comments to authors, only to editors.

Response: Thank you.